# Stress Corrosion Cracking Behavior of Alloy 600 Coupled to Magnetite under High-Temperature Caustic Conditions

**DOI:** 10.3390/ma12132091

**Published:** 2019-06-28

**Authors:** Geun Dong Song, Jeoh Han, Soon-Hyeok Jeon, Do Haeng Hur

**Affiliations:** Korea Atomic Energy Research Institute, 989-111 Deadeok-daero, Yuseong-gu, Daejeon 34057, Korea

**Keywords:** Nickel alloys, steam generator, stress corrosion cracking, magnetite, galvanic couple

## Abstract

This study aims to investigate and explain the magnetite-accelerated stress corrosion cracking phenomenon of Alloy 600 under caustic conditions, based on the electrochemical behavior. After the SCC test that lasted for 300 h, no cracks were observed in any of the magnetite-free specimens, whereas cracks with a depth of 150 to 280 μm were generated in all the magnetite-deposited specimens. Furthermore, the electrochemical behavior of magnetite and Alloy 600 demonstrated that Alloy 600 behaved as an anode in the coupling system with magnetite. In this coupling system, the electrochemical potential of Alloy 600 can be shifted into the range potentially susceptible to stress corrosion cracking.

## 1. Introduction

Heat-transfer tubing made of nickel-based Alloy 600 has undergone a variety of corrosion-related damages in the secondary environments of steam generators (SG) in pressurized water reactors (PWRs): stress corrosion cracking (SCC); intergranular corrosion; wastage; pitting [1,2]. Because of these corrosion-related problems, some SGs with a design life of forty years have been replaced after only about ten years of operation [3]. As a result, Alloy 600 has been replaced by heat-treated Alloy 690 that contains about 30 wt% Cr.

SCC is well known as the major mechanism of Alloy 600 SG tubing failure occurring in the secondary environment of PWRs. This degradation mode mainly occurs in heated crevices covered with porous magnetite deposits, because a severe corrosion environment can be formed in these crevices as a result of high impurity concentrations caused by local boiling [4,5,6,7]. Extensive laboratory-scale investigations related to SCC of Alloy 600 have shown that water chemistry factors, such as the concentration of harmful impurities and pH changes, can promote the SCC of Alloy 600 [8,9,10,11,12]. However, even if a corrosive environment can be formed in the heated crevices covered with magnetite, this is still not enough to explain the extremely rapid SCC of Alloy 600 experienced in the field.

The reason for the occurrence of rapid SCC failure in Alloy 600 remains a problem that must be solved in this field. To explain the root causes for this, the fact that the real corrosion-related degradation is generated on the porous magnetite-deposited tube surface needs to be considered carefully. Specifically, only the tube surface connected through the pores inside porous magnetite deposits is locally exposed to the crevice environment. At the same time, the tube surface is in electrical contact with the magnetite deposit. Magnetite is well known as a conductive material, and its electrical conductivity indicates almost metallic behavior [13,14]. Consequently, a galvanic cell can be activated on the tube surface in contact with magnetite. In this regard, it was reported that SCC can be promoted owing to the enhanced electrochemical kinetics of the external surface by the environmental factors, such as pH, oxygen concentration, flow rate, temperature, and conductivity of electrolyte [15,16]. In addition, the results have shown that the crack growth rate can be accelerated when the electrochemical potential of materials is raised toward the anodic direction [16]. Thus, if the tube surface behaves as an anodic member in the coupling system with magnetite, this may become a potential factor that accelerates SCC. Therefore, the goal of the work described in this communication was to investigate SCC and electrochemical properties of Alloy 600, coupled to magnetite under a crevice environment. For this, a specially designed magnetite-deposited SCC specimen was prepared, and the effect of magnetite on the electrochemical behavior was investigated.

## 2. Experimental Methods

### 2.1. SCC Test

An Alloy 600 ingot was manufactured using a vacuum furnace equipped with a high-frequency induction heating system (Carbolite, Hope valley, UK) and was formed into a plate by hot-rolling in the temperature range of 1050 to 1150 °C. The plate thickness was finally reduced to about 1.5 mm through a cold-rolling process. The cold-worked plate was machined into specimens with dimensions shown on the left side of Figure 1. After that, the machined specimens were annealed at 1060 ^°^C for 150 s to dissolve the carbon in the matrix and control the microstructure (average grain size: about 27 μm), and then quenched in water at room temperature. These specimens were abraded using SiC paper up to #1000 grit, and then fabricated into a U shape (the right side of Figure 1) for SCC tests according to the guideline provided by ASTM G30-97.

To investigate SCC behavior caused by the coupling effect of Alloy 600 and magnetite, we needed to prepare SCC specimens electrically connected with magnetite. To achieve this, a thick magnetite film was grown on the entire surface of the prepared SCC specimen using the electrodeposition (ED) method, except for an area with a width of approximately 800 μm in the apex region (the left side in Figure 2). The ED of the magnetite film was carried out in a solution containing 2.0 M NaOH, 0.043 M Fe_2_(SO_4_)_3_, and 0.1 M triethanolamine by using a three-electrode system under the following conditions: ED potential = −1.05 V_SCE_; ED time = 120 min; ED temperature = 80 ^°^C. The details of the ED process and characteristics of the magnetite film are given in our previous reports [17,18,19].

The right side of Figure 2 shows the final product of the magnetite-deposited SCC specimen after the ED process. In this specimen, the undeposited area corresponds to the local tube surface connected through the pores inside porous magnetite deposits. Consequently, a galvanic cell can be consistently activated at the undeposited area/magnetite film interface, as shown in the 3D optical microscope image on the bottom of the final product image in Figure 2. 

The SCC tests for the magnetite-free and magnetite-deposited specimens were carried out separately in deaerated 10 wt% NaOH solutions for 300 h at 315 ^○^C by using a static Ni-autoclave with a capacity of 2 L. This test condition was chosen to simulate the crevice environment of the secondary side in the SG of PWRs. After the SCC test was completed, all specimens were cut along the center lengthwise to examine the stress corrosion cracks. These samples were embedded in a cold mounting resin, polished with 0.3 μm Al_2_O_3_ powder, and observed using an optical microscope (OM) (VHX-6000; Keyence, Osaka, Japan).

### 2.2. Potentiodynamic Polarization Test

The Alloy 600 specimen for the potentiodynamic polarization test was machined into a rectangular shape (6 mm × 6 mm × 1.5 mm) and was spot-welded to an Alloy 600 wire that had been covered with a heat-shrinkable polytetrafluoroethylene (PTFE) tube. This specimen was used as an Alloy 600 electrode. A magnetite electrode was prepared by ED of a magnetite film on the entire surface of the prepared Alloy 600 electrode under the same conditions as had been used for the preparation of the magnetite-deposited SCC specimen.

The potentiodynamic polarization test was performed in the same solution used for the SCC test at 280 ^°^C by using a static Ni-autoclave with a capacity of 1 L shown in Figure 3. An external Ag/AgCl electrode with 0.01 N KCl was used as the reference electrode, and a platinum wire was used as the counter electrode. After the open-circuit potential (OCP) was stabilized at 280 ^°^C, polarization scans were performed from the OCP toward the positive or negative direction at a scan rate of 30 mV/min. 

The measured potentials were converted to the standard hydrogen electrode (SHE) scale according to the following relationship [20,21]:(1)$$E_{\mathrm{SHE}}=E_{\mathrm{obs}}+0.3432\;-0.001\left(T-t\right)+5.4081\;\times \;10^{-7}{\left(T-t\right)}^{2}\;-5.4906\times 10^{-9}{\left(T-t\right)}^{3},$$
where E_SHE_ represents the electrode potential versus SHE and E_obs_ is the measured electrode potential. T is the experimental temperature, and t is the room temperature (25 ^°^C). In addition, the reproducibility of the polarization curve was confirmed by repeatedly performing the tests at least three times using newly prepared electrodes and test solutions.

## 3. Results and Discussion

Figure 4 shows the cross-sectional OM image of the U-bend specimens exposed to 10 wt% NaOH solution for 300 h. No stress corrosion cracks were generated in any of the magnetite-free specimens. However, stress corrosion cracks with a depth of 150 to 280 μm were generated in all magnetite-deposited specimens. Furthermore, the deepest crack was always observed near the junction between the exposed Alloy 600 surface and magnetite layer, not the apex with the greatest stress in the U-bend specimen. These results indicate that magnetite coupled to Alloy 600 induces the rapid SCC of Alloy 600.

Figure 5 shows the anodic polarization curve of Alloy 600 and cathodic polarization curve of magnetite. The corrosion potential of magnetite was approximately 100 mV higher than that of Alloy 600. This implies that Alloy 600 behaves as an anodic member of the galvanic couple with magnetite if these two materials are in electrical contact. In this couple with equal areas of the materials, the corrosion potential of Alloy 600 is expected to be shifted to a more positive value. In addition, the anodic current of Alloy 600 will be increased by galvanic coupling with magnetite. To evaluate the effect of the increased magnetite surface area in the couple, the cathodic curve of magnetite with an area of 50 cm^2^ was also presented in Figure 5, which has been simply estimated from that with an area of 1 cm^2^. The corrosion potential of the coupled Alloy 600 will be more polarized to the active–passive transition region with increasing the magnetite surface area, resulting in the greatly increased anodic current. Therefore, it is considered that these changes in the electrochemical behavior also occur on the exposed Alloy 600 surface in the magnetite-deposited specimen shown in Figure 4 during the SCC test. 

Particularly, the alloy which exhibits an active–passive behavior in the anodic polarization curve is known to be more susceptible to SCC in its active–passive transition region [22]. In this regard, Pessall et al. and subsequent researchers reported that the caustic SCC of Alloy 600 was significantly accelerated when the electrochemical potential of Alloy 600 was held in the potential range corresponding to the active–passive transition region of the anodic polarization curve [23,24,25]. Consequently, the rapid SCC phenomenon in the magnetite-deposited specimen shown in Figure 4 is because the electrochemical potential of Alloy 600 is shifted to the more susceptible potential region to SCC by the galvanic corrosion mechanism with magnetite (shown in Figure 5). In addition, this changes in the electrochemical behavior can affect the formation of the oxide layer on Alloy 600. In this regard, we have reported recently that the magnetite coupled to Alloy 600 caused the fast growth of oxide film in water containing lead oxide at 315 °C, resulting in the formation of relatively Cr-depleted inner oxide layers with defects [26]. Furthermore, the result has shown that SCC was accelerated by this defective nature in oxide layer properties. Therefore, under high-temperature caustic conditions, it is also expected that the oxide layer formed on the magnetite-deposited specimen has to be more defective than that formed on the magnetite-free specimen.

Many investigations related to the SCC phenomena of SG tubing occurring in crevices covered with magnetite deposit have mainly focused on water chemistry factors, such as pH, temperature, and impurity concentration. However, it should be noted that SG tubes are always corroded under the situation which is in contact with the porous magnetite deposit in the secondary side of PWRs. Especially, in the crevice region between the SG tubes and their support plate, flow-induced vibration can lead to tube damage by fretting-wear due to impact and sliding movement between the tubes and their support plates [2,27,28]. This wear process interferes the formation of passive film on the surface of SG tubes, and as a result, the bare surface of SG tube is exposed consistently to a corrosive crevice environment, in which aggressive impurities are concentrated. Furthermore, galvanic cells on the wearied surface of SG tube can be activated, because the crevice is covered with porous magnetite deposits in operating SGs. Based on the result of potentiodynamic polarization tests in Figure 5, the electrochemical potential of Alloy 600 can be shifted to active–passive transition region of the anodic polarization by the galvanic corrosion mechanism with magnetite, resulting in the significantly increased corrosion current. In this potential region, re-passivation of the exposed alloy surface by the fretting-wear can be retarded, because the state of the passive film of an alloy in the active–passive region is unstable and weak [22]. Therefore, the Alloy 600 SG tube can be more susceptible to SCC in this potential range [29]. The result shown in Figure 4 demonstrates clearly that magnetite coupled to Alloy 600 induced a fast SCC of Alloy 600. Consequently, the combined effects of fretting-wear, the impurity concentrations, and the galvanic coupling with magnetite can cause the fast SCC of Alloy 600 in the field.

This work confirms that the caustic SCC of Alloy 600 is significantly accelerated when the alloy is in electrical contact with magnetite. From the perspective of electrochemical behavior, it is demonstrated for the first time that the galvanic coupling with magnetite can shift the electrochemical potential of Alloy 600 into the more susceptible potential region to SCC. Therefore, the magnetite-accelerated stress corrosion cracking (MASCC) mechanism should be considered as a new acceleration factor of SCC in Alloy 600 SG tubes in secondary side conditions, in addition to the impurity concentration within the porous magnetite deposits.

## 4. Conclusions

The SCC behavior of Alloy 600, coupled with magnetite, has been investigated under a high-temperature caustic condition. Magnetite in contact with Alloy 600 induced rapid SCC of Alloy 600. Regarding electrochemical behavior, Alloy 600 and magnetite acted as an anodic and cathodic members, respectively, in the coupling system. In this couple, the electrochemical potential of Alloy 600 was shifted into the active–passive transition region of the anodic polarization curve, which is the range susceptible to SCC. Based on the experimental results, a galvanic corrosion mechanism between magnetite and Alloy 600 is proposed as a new acceleration contributor to a rapid cracking of Alloy 600 in the PWR secondary environment.

## Figures and Tables

**Figure 1 materials-12-02091-f001:**
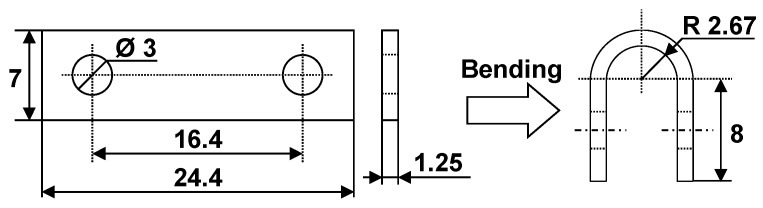
Dimensions of a U-bend specimen for the stress corrosion cracking (SCC) tests (unit: mm).

**Figure 2 materials-12-02091-f002:**
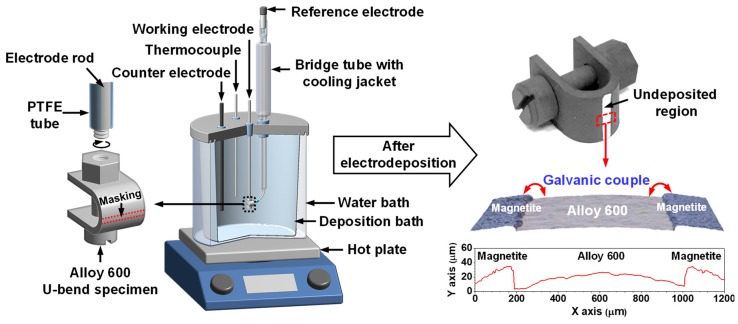
Schematic for the preparation of the magnetite-deposited SCC specimen.

**Figure 3 materials-12-02091-f003:**
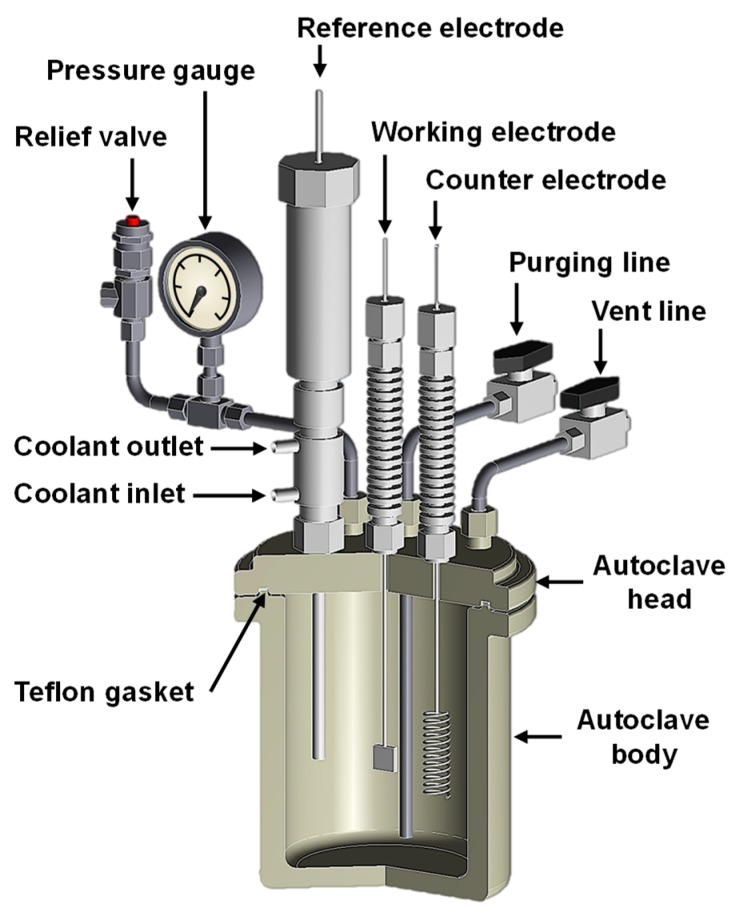
Schematic for the preparation of the magnetite-deposited SCC specimen.

**Figure 4 materials-12-02091-f004:**
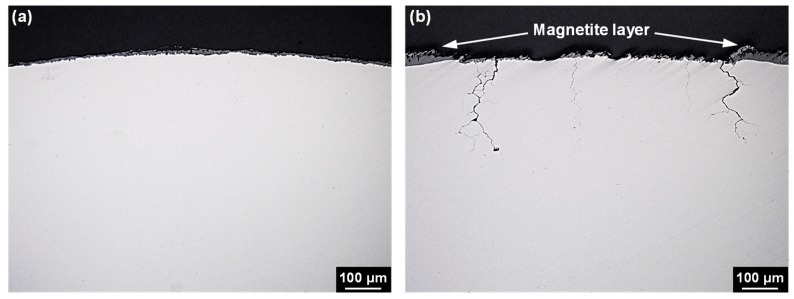
Cross-sectional optical microscope (OM) images of the Alloy 600 SCC specimens after the SCC tests: (**a**) magnetite-free SCC specimen and (**b**) magnetite-deposited SCC specimen.

**Figure 5 materials-12-02091-f005:**
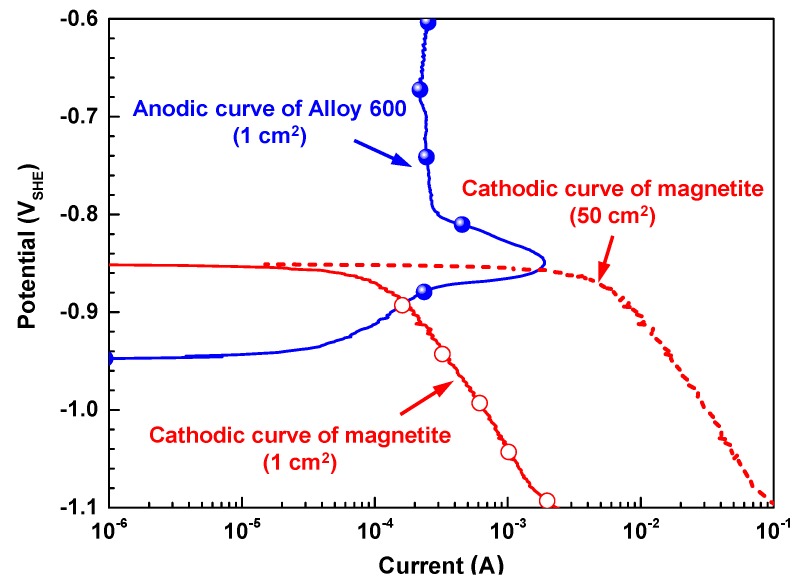
Electrochemical behavior of Alloy 600 and magnetite in 10 wt% NaOH solution at 280 ^°^C.

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
