# Peer review of "Stress Corrosion Cracking Behavior of Alloy 600 Coupled to Magnetite under High-Temperature Caustic Conditions"

_materials, 2019, doi:10.3390/ma12132091_

Round 1
Reviewer 1 Report
The authors present a nice work on "Stress Corrosion Cracking Behavior of Alloy 600 Coupled to Magnetite under High-Temperature 3 Caustic Conditions". They describe the introduction, experiment, results and discussion well organized. They also find something new in stress corrosion cracking of Alloy 600. However, there is a minor correction needed in Figure 5. The unit of potential should be added!
Author Response
Dear Editor and Reviewers.
I have revised the manuscript entitled “Stress Corrosion Cracking Behavior of Alloy 600 Coupled to Magnetite under High-Temperature Caustic Conditions”. (Manuscript No.: Materials-512691).
Thanks to the reviewer’s kind reviews and comments, we had a valuable chance to improve the quality of our manuscript. We did our best to address the each comment, although some of them seemed to be a little bit insufficient.
The revised parts of the manuscript are highlighted in yellow so that you can easily view the change. Please find the author’s response to the reviewer’s comments on the attached PDF file, where we summarized our answers to each of the comments.
I sincerely hope the revised version is up to the reviewer’s expectations.
Yours sincerely,
Do Haeng Hur
Korea Atomic Energy Research Institute

Reviewer 2 Report
This manuscript by Song et al. reported the accelerated SCC of an Alloy 600 in PWR secondary water environment when the alloy was in electrical contact with a magnetite layer. The experimental results and discussion, by themselves, may be technically correct. However, it is widely known that when Alloy 600 is subjected to high-temperature water environments, the resultant surface oxides are composed of two layers, i.e., an inner layer of Cr-rich spinel and an outer layer of Fe/Ni-rich spinel. While the outer layer may be porous, the inner layer is dense and resistant to ion diffusion. Thus, it is essentially doubtful if the coupling system designed in this work, in which the magnetite layer was directly deposited onto the alloy, credibly reflects the electrochemical environment at which SCC is initiated under service. Thus, to establish the role of magnetite layer, the authors are prompted to provide evidence regarding the local state of specific section of the tube surface if it is considered more susceptible to SCC. For instance, is the inner layer inhomogeneously formed or selectively broken at the site of SCC?
Author Response

(The authors gave the same response as above.)

Reviewer 3 Report
This is a very good paper, but the authors fail to recognize that their finding, that coupling to magnetite results in enhanced SCC, were predicted by the Coupled Environment Fracture Model (CEFM) several decades ago, because of the enhanced kinetics of the cathodic partial reaction on the external surface (magnetite), as shown by their Figure 5. The authors should read, amend the paper accordingly, and cite: “Theoretical Estimation of Crack Growth Rates in Type 304 Stainless Steel in BWR Coolant Environments”. Corrosion, 52(10), 768-785 (1996) and “On the Shape of Stress Corrosion Cracks in Sensitized Type 304 SS in Boiling Water Reactor Primary Coolant Piping at 288 °C”, J. Nucl. Mater., 454(1-3), 359-372 (2014). Please note that this is a mandatory correction.
Author Response

(The authors gave the same response as above.)

Round 2
Reviewer 2 Report
It can be seen that the authors attempted to address the question about their experimental design by referring to their early work on the SCC of Alloy 600 in a slightly different water environment, which was suggested to be accelerated by the magnetite layer. Their published work should be respected. However, it remains doubtful if the real environment at the metal-water interface can be simulated with the magnetite layer in direct contact with the alloy.
It should be noticed that, as long as the alloy is subjected to oxidation in HT water, the inner and outer oxides are formed simultaneously. The outer layer may subsequently become porous, but the inner layer should always present underneath. Thus, it is hard to imagine the magnetite is directly connected to the alloy. If this is ever true for the authors' studied material-environment pairing, they are prompted to provide direct evidence or appropriate references regarding a) the completeness of coverage, or b) the electrical conductivity, of the inner layer in the real (rather than their simulated) system.
Author Response
Dear Reviewer.
Thanks to the your valuable comments, we had a valuable chance to improve the quality of our manuscript. We did our best to address the each comment.
The revised parts of the manuscript are highlighted in yellow so that you can easily view the change. Please find the author’s response to the comments on the attached PDF file, where we summarized our answer to the comments.
I sincerely hope the revised version is up to the reviewer’s expectations.
Yours sincerely,
Do Haeng Hur
Korea Atomic Energy Research Institute

Reviewer 3 Report
The missing previous work has now been cited.
Author Response
Dear Reviewer.
Thanks to your kind comments, we had a valuable chance to improve the quality of our manuscript.
Yours sincerely,
Do Haeng Hur
Korea Atomic Energy Research Institute